# DeepShop: A Benchmark for Deep Research Shopping Agents

## Abstract

Web agents for online shopping have shown great promise in automating user interactions across e-commerce platforms. Benchmarks for assessing such agents do not reflect the complexity of real-world shopping scenarios, as they often consist of overly simple queries with deterministic paths, such as "Find iPhone 15." Real shopping scenarios are inherently more layered, involving multi-dimensional product attributes, search filters, and user-specific sorting preferences. To address this gap, we introduce DeepShop, a benchmark designed to evaluate web agents in complex and realistic online shopping environments. DeepShop comprises three key components. (1) Query diversity evolution: Starting from real user queries, we generate diverse queries across five popular online shopping domains. (2) Query complexity evolution: We further evolve these queries to increase complexity, considering product attributes, search filters, and sorting preferences, and classify them into three levels: easy, medium, and hard, based on the number of evolutions. (3) Fine-grained and holistic evaluation: We propose an automated evaluation framework that assesses agent performance in terms of fine-grained aspects (product attributes, search filters, and sorting preferences) and reports the overall success rate through holistic evaluation. We conduct a systematic evaluation of retrieval-augmented generation (RAG) methods, web agents, and deep research systems. Results show that RAG struggles with complex queries due to its lack of web interaction, while other methods face significant challenges with filters and sorting preferences, leading to low overall success rates. We also perform cross-category, complexity-based evaluations and error analyses to support the advancement of deep research shopping agents.

## 1 Introduction

Recent progress in web agents has enabled more complex automation of human interactions on e-commerce platforms (Yao et al., 2022; Kim et al., 2023; Sumers et al., 2024), largely driven by the integration of large language models (LLMs) that provide planning, memory, and web interaction capabilities (He et al., 2024a; Sumers et al., 2024; Zheng et al., 2024). Despite these advances, web agents still face significant challenges in completing complex user queries in dynamic, real-world shopping environments (Nguyen et al., 2024; Zhang et al., 2024a). These complex user queries require agents to perform deep research of e-commerce platforms—browsing product listings, applying filters, and comparing items—to accommodate diverse and nuanced user preferences (Sondhi et al., 2018; Chen et al., 2024b). This leads to our key research question: *Can existing web agents effectively fulfill diverse and complex user needs in realistic shopping scenarios?*

To evaluate web shopping agents, recent studies have proposed benchmarks that test their ability to complete user tasks via simulated or real website interactions. Most offline benchmarks, such as Mind2Web (Deng et al., 2023), WebShop (Yao et al., 2022), and WebArena (Zhou et al., 2024), are based on static environments constructed from pre-collected web snapshots or manually curated HTML structures. While these benchmarks enable controlled evaluations, they fail to capture the dynamic and unpredictable nature of real-world websites, which often feature noisy, frequently updated, and interactive content (Wang et al., 2024; Ning et al., 2025).

Recently, several online benchmarks have emerged, including Mind2Web-Live (Pan et al., 2024) and WebVoyager (He et al., 2024a), which enable agents to operate within real-time web environments.

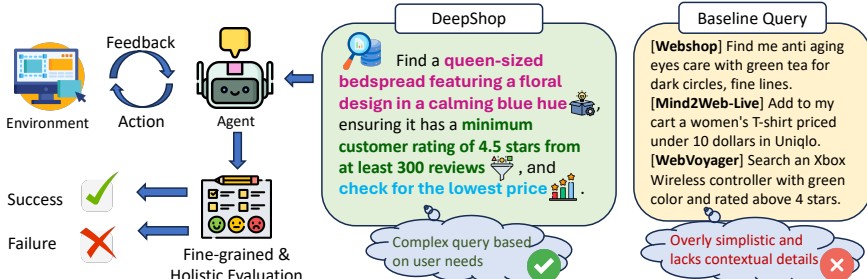

Figure 1: DeepShop evaluates agents on realistic and complex shopping queries with fine-grained, holistic metrics, while existing benchmarks use overly simple queries lacking contextual depth.

While recent efforts mark progress, they still fail to capture the complexity and diversity of real-world shopping (Song et al., 2025; Xue et al., 2025), as most benchmark tasks remain simple and deterministic (e.g., "find an iPhone 15"), unlike real queries that demand multi-attribute reasoning, filtering, and personalized sorting (Nguyen et al., 2024; Zhang et al., 2024a).

To bridge this gap, we introduce DeepShop, a benchmark specifically designed to evaluate web agents in complex online shopping scenarios. DeepShop is tailored to evaluate web shopping agents in handling diverse and complex user queries, and includes a comprehensive evaluation framework. The key components of DeepShop are as follows:

- **Query diversity evolution**: We begin with real-world user shopping queries and generate a wide range of goals across five popular product categories (Books, Electronics, Home, Fashion and Sports). This ensures that agents must generalize across varied user shopping intents.
- **Query complexity evolution**: We progressively enhance the complexity of each query by introducing combinations of product attributes (e.g., brand, color), search filters (e.g., ratings, availability), and sorting preferences (e.g., lowest price first). These queries are categorized into *easy*, *medium*, and *hard* levels, based on the number and type of components involved.
- **Fine-grained and holistic evaluation**: To enable meaningful comparisons across agents, we design an automated evaluation pipeline that assesses performance on three axes—correct attribute matching, correct use of filters, and proper execution of sorting preferences—alongside a holistic success rate measuring task completion.

We conduct both fine-grained and holistic evaluations of various approaches, including simple retrieval-augmented generation (RAG) methods, advanced web agents, and commercial deep research systems. Results show that RAG methods, lacking web interaction, struggle with DeepShop queries; web agents fail to handle filters and sorting well; and even deep research systems fall short on filtering, leading to low overall success rates. We further analyze performance across product categories, query complexities, and error types to guide future research. By introducing a benchmark that mirrors real-world complexity, DeepShop provides a rich testbed for advancing agent planning, adaptability, and generalization, bridging the gap between academic systems and real-world deployment.

The contributions of this paper are as follows:

- We present DeepShop, a comprehensive benchmark for evaluating web agents in complex online shopping scenarios, featuring diverse queries across five product categories and varying complexity levels. Our dataset is built through a multi-stage process that evolves real-world shopping intents by expanding query diversity and complexity.
- We conduct extensive experiments comparing simple RAG methods, advanced web agents, and commercial deep research systems using our fine-grained and holistic evaluation framework.
- We provide detailed analyses across product categories, query complexity levels, and specific error types, revealing critical limitations in current systems and offering insights to guide future development of more effective deep research shopping agents.

## 2 RELATED WORK

**Benchmarks for web agent evaluation.** Existing benchmarks for web agent evaluation fall into two categories: offline and online, as shown in Table 1. Offline benchmarks (e.g., Mind2Web (Deng et al.,

Table 1: Comparison of existing benchmarks and DeepShop. DeepShop is evaluated online across diverse product categories, providing fine-grained assessment over product attributes, search filters, and sorting preferences. Average token length is computed from 100 randomly sampled queries.

| Benchmark | Avg. query length | Env. type | Product category | Product attribute | Search filter | Sorting preference | Task success |
|---|---|---|---|---|---|---|---|
| Webshop (Yao et al., 2022) | 18.2 | Offline | ✓ | ✓ | ✗ | ✗ | ✓ |
| Mind2Web (Deng et al., 2023) | 13.4 | Offline | ✗ | ✗ | ✗ | ✗ | ✓ |
| Webarena (Zhou et al., 2024) | 19.9 | Offline | ✗ | ✗ | ✗ | ✗ | ✓ |
| VWebarena (Koh et al.) | 21.4 | Offline | ✗ | ✗ | ✗ | ✗ | ✓ |
| MMInA (Zhang et al., 2024c) | 24.2 | Offline | ✗ | ✗ | ✗ | ✗ | ✓ |
| ChatShop (Chen et al., 2024a) | 20.4 | Offline | ✗ | ✗ | ✗ | ✗ | ✓ |
| WebLINX (Lù et al., 2024) | 6.2 | Online | ✗ | ✗ | ✗ | ✗ | ✓ |
| Mind2Web-Live (Pan et al., 2024) | 16.2 | Online | ✗ | ✗ | ✗ | ✗ | ✓ |
| WebVoyager (He et al., 2024a) | 29.5 | Online | ✗ | ✗ | ✗ | ✗ | ✓ |
| **DeepShop (Ours)** | 62.0 | Online | ✓ | ✓ | ✓ | ✓ | ✓ |

2023), WebShop (Yao et al., 2022), WebArena (Zhou et al., 2024), ChatShop (Chen et al., 2024a)) use static snapshots or simulated environments, offering controlled conditions but failing to capture the dynamic nature of real-world websites (Koh et al.; Jang et al., 2024). In contrast, online benchmarks (e.g., WebVoyager (He et al., 2024a), Mind2Web-Live (Pan et al., 2024)) provide realistic real-time settings but focus on general and simple tasks, leaving complex web shopping queries underexplored. Even basic RAG systems with LLMs and Google Search can perform strongly on many current benchmarks (Yoran et al., 2024; Mialon et al., 2024). To address this gap, DeepShop introduces a benchmark targeting challenging online web shopping queries, constructed by query diversity and complexity evolution. We also propose fine-grained evaluation metrics across product attributes, search filters, and sorting preferences to offer a comprehensive assessment of agent performance.

**Web agents for task automation.** Recent progress in web agents has followed a clear trajectory, evolving from text-based to multimodal systems. Early HTML-based agents, such as WebGPT (Nakano et al., 2021), MindAct (Deng et al., 2023), and Agent-E (Abuelsaad et al., 2024), leverage LLMs to interpret language instructions and navigate web interfaces using DOM trees (Gur et al., 2024; Lai et al., 2024). Building on this, multimodal, vision-based agents like SeeAct (Zheng et al., 2024), WebVoyager (He et al., 2024a), and Browser Use (Müller & Žunič, 2024) integrate visual grounding to handle complex layouts and interactive components (Shaw et al., 2023; Furuta et al., 2024). Recent systems such as OpenAI Deep Research (OpenAI, 2025) and Gemini Deep Research (Gemini, 2025) use advanced reasoning LLMs to tackle complex information-seeking tasks. Despite these advances, most evaluations remain limited to generic benchmarks, leaving agent performance on complex, real-world shopping scenarios underexplored. In this paper, we evaluate simple RAG methods, text-based and multimodal web agents, and deep research systems in realistic online shopping environments.

**Query understanding in E-commerce.** Online shopping platforms have become central to modern consumer behavior, making accurate query understanding critical for satisfying user experiences (Hirsch et al., 2020; Zhang et al., 2020; Ren et al., 2024). However, many e-commerce queries involve overwhelming product spaces and complex user preferences that are difficult to express with simple keywords or filters (Sondhi et al., 2018). Traditional information retrieval (IR) systems often struggle with such complexity (Tsagkias et al., 2020; Chen et al., 2024b), while conversational IR systems, despite supporting multi-turn preference elicitation, remain constrained by training products and cannot autonomously browse web content (Zhang et al., 2024b; Chen et al., 2024a). Recent advances in web agents offer a promising alternative by autonomously interacting with e-commerce sites, searching for relevant items, and mimicking human browsing behaviors (Yao et al., 2022; He et al., 2024a). To advance this line of research, we introduce DeepShop, a benchmark designed to evaluate web agents on complex e-commerce queries and drive progress in web shopping automation.

## 3 DEEPSHOP BENCHMARK

In this section, we present the DeepShop benchmark, a framework for evaluating web agents in realistic online shopping environments for complex user queries. We begin by formulating the web

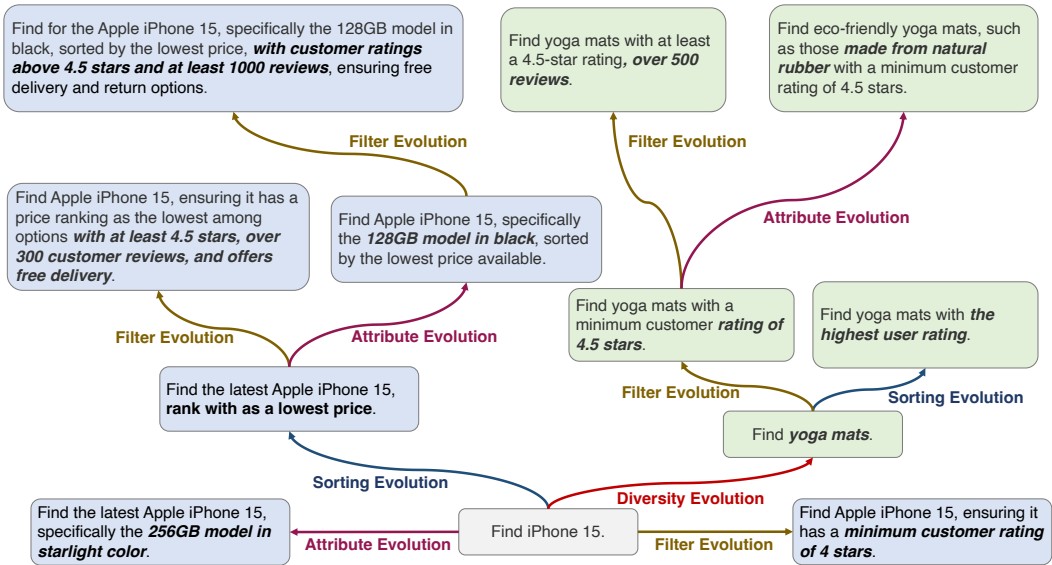

Figure 2: Running examples of diversity and complexity evolution in DeepShop. Complexity evolution includes attribute evolution, filter evolution, and sorting evolution.

shopping tasks. Next, we describe the processes of query diversity and complexity evolution, which are derived from real user seed queries, as shown in Figure 2. We then analyze key characteristics of DeepShop to offer a deeper understanding of the dataset. Finally, we introduce a comprehensive evaluation framework that incorporates both fine-grained and holistic metrics.

## 3.1 TASK FORMULATION

Following previous work (He et al., 2024a; Pan et al., 2024), we formulate the online web shopping task as a partially observable Markov decision process (POMDP) (Kaelbling et al., 1998) defined by a tuple $(\mathcal{S}, \mathcal{O}, \mathcal{A}, \mathcal{T})$, where $\mathcal{S}$ denotes the state space, $\mathcal{O}$ the observation space, $\mathcal{A}$ the action space, and transition function $\mathcal{T} : \mathcal{S} \times \mathcal{A} \to \mathcal{S}$. In this setting, at each time step $t$, given a user query $q$, the web shopping agent receives an observation $o_t \in \mathcal{O}$ that partially reflects the underlying state $s_t \in \mathcal{S}$ of the web environment. The agent then takes an action $a_t \in \mathcal{A}$, resulting in a new environment state $s_{t+1} \sim \mathcal{T}(s_t, a_t)$ and an updated observation $o_{t+1} \in \mathcal{O}$.

## 3.2 SEED DATA CURATION

To evaluate web agents under realistic user shopping intentions, we curate a seed dataset by selecting a subset of web shopping queries from two real-world benchmarks: Mind2Web-Live (Pan et al., 2024) and WebVoyager (He et al., 2024a). Specifically, we manually select 50 user queries and categorize them into five representative shopping domains: *Books* (4), *Electronics* (14), *Home* (20), *Fashion* (5), and *Sports* (7). We define each domain as follows:

- **Books** $d_{\text{books}}$: Physical books, eBooks, and audiobooks across various genres.
- **Electronics** $d_{\text{electronics}}$: Electronics and digital products such as smartphones, tablets, laptops, headphones, and smart devices.
- **Home** $d_{\text{home}}$: Household items including furniture, appliances, cleaning tools, and daily necessities.
- **Fashion** $d_{\text{fashion}}$: Apparel, footwear, and accessories for all genders and age groups.
- **Sports** $d_{\text{sports}}$: Fitness and recreational equipment, sportswear, and training accessories.

## 3.3 SHOPPING QUERY DIVERSITY EVOLUTION

Existing web shopping datasets (Yao et al., 2022; He et al., 2024a; Pan et al., 2024; Deng et al., 2023; Koh et al.; Jang et al., 2024) often overlook fine-grained product categories, limiting their overall diversity. To address this limitation, inspired by Xu et al. (2023), we generate entirely new queries based on the original query and a randomly selected product category through the following diversity

evolution process:

$$q_i^* = \text{Diversity}(q_i, d), \tag{1}$$

where $\text{Diversity}(\cdot)$ is implemented by prompting GPT4-o models, $q_i \in \mathcal{D}_{\text{original}}$ is a seed query, and $d \in \{d_{\text{books}}, d_{\text{electronics}}, d_{\text{home}}, d_{\text{fashion}}, d_{\text{sports}}\}$ denotes a randomly selected product category. Finally, we construct the web shopping diversity evolution dataset $\mathcal{D}_{\text{diversity}}$ by combining the seed dataset with all generated queries: $\mathcal{D}_{\text{diversity}} = \mathcal{D}_{\text{original}} \cup \{q_i^*\}_{i=1}^N$, where $N$ is the number of seed queries.

### 3.4 SHOPPING QUERY COMPLEXITY EVOLUTION

To increase the complexity of web shopping queries, we propose a web shopping complexity evolution strategy. Specifically, we focus on three key augmentation directions: (1) *Product attributes*, referring to concrete product characteristics users may specify to express detailed intent; (2) *Search filters*, representing categorical or numerical constraints commonly used on e-commerce platforms; and (3) *Sorting preferences*, indicating desired result orderings, such as price or popularity. Specifically, we perform iterative complexity evolution to progressively enhance query complexity. In each iteration $t$, one of the three strategies is randomly selected to evolve the query $q_{i,t}$ from the previous step:

$$q_{i,t+1} = \text{Complexity}(q_{i,t}, c), \tag{2}$$

where $\text{Complexity}(\cdot)$ is implemented by prompting GPT4-o, $q_{i,t}$ denotes the $i$-th query in $t$-th complexity evolution, $i \in [1, |\mathcal{D}_{\text{diversity}}|]$, $t \in [1, T]$, $q_{i,0}$ denotes the $i$-th query from $\mathcal{D}_{\text{diversity}}$, and $c \in \{c_{\text{attr}}, c_{\text{filter}}, c_{\text{sort}}\}$ is the randomly selected strategy. The three complexity evolution strategies are summarized as follows:

- **Attribute evolution $c_{\text{attr}}$**: Enhance the query by incorporating concrete product attributes, such as *brand*, *model*, *price range*, *color*, *size*, *weight*, or unique features of products.
- **Filter evolution $c_{\text{filter}}$**: Enhance the query by adding specific search filter commonly available on e-commerce platforms. These include constraints like *minimum customer rating* (e.g., 4.5 stars), *minimum number of reviews* (e.g., 500+), *shipping options* (e.g., free delivery), *release timeframe* (e.g., new arrivals in the past 30 days), *return policies*, or *warranty information*.
- **Sorting evolution $c_{\text{sort}}$**: Enhance the query by appending a sorting preference, directing the system to find top-ranked products according to criteria like *lowest price*, *highest user rating*, *newest arrival*, or *best seller ranking*.

By iteratively applying the above strategies, our method mimics the natural evolution of user queries, generating a hierarchical set of increasingly complex queries. Starting from diverse queries in $\mathcal{D}_{\text{diversity}}$, we apply $T = 5$ rounds of complexity evolution, resulting in a total of 600 queries.

### 3.5 DATASET ANALYSIS

**Analysis of query diversity evolution.** Existing benchmarks for online web shopping often exhibit skewed distributions across product categories, introducing evaluation bias and limiting the generalizability of agent performance, as shown in Figure 3. To mitigate this, we construct a balanced subset of 150 queries from our 600-query pool, systematically selecting 30 queries each from five major categories: Books, Electronics, Home, Fashion, and Sports. Following previous work (He et al., 2024a;b), we manually verify each generated task and, if necessary, revise it to ensure high quality and confirm that the answers are available on the corresponding website. While this uniform category distribution does not necessarily reflect real-world query frequency, it provides a controlled and equitable test bed for evaluating cross-domain generalization. The DeepShop benchmark significantly reduces the category imbalance present in the seed data, enabling more controlled and equitable comparisons. This balanced design helps isolate category-related performance effects, offering a clearer assessment of an agent's ability to generalize beyond narrow domain specialization.

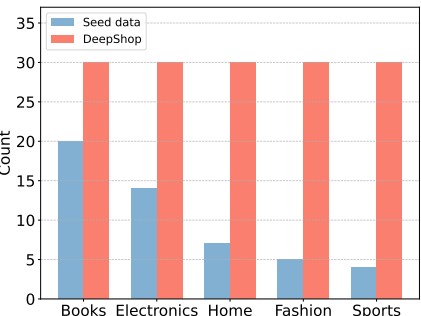

Figure 3: Product category distribution after query diversity evolution.

**Analysis of query complexity evolution**. The complexity evolution strategy progressively enhances query complexity by incorporating additional product attributes, search filters, or sorting preferences.

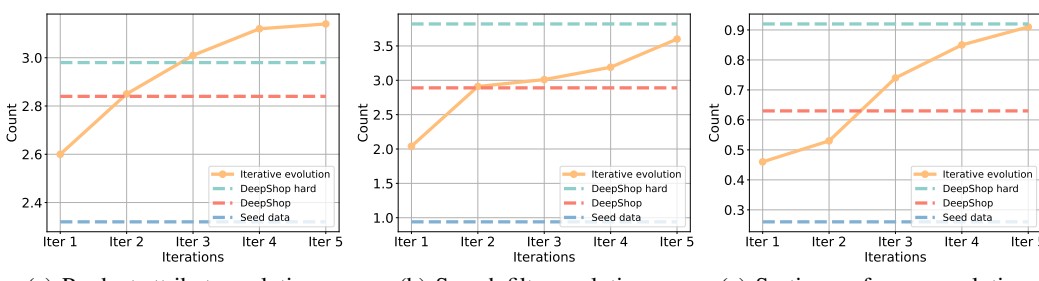

(a) Product attribute evolution.    (b) Search filter evolution.    (c) Sorting preference evolution.

Figure 4: Analysis of query complexity evolution.

We perform a fine-grained analysis of query evolution across these three dimensions. Regarding product attributes, as depicted in Figure 4(a), the average number of product attributes per query exhibits a steady increase throughout the iterations. Ultimately, the average number of product attributes in DeepShop surpasses the seed data by 0.52, while the hard subset contains an additional 0.66 attributes on average. In terms of search filters, Figure 4(b) illustrates a consistent increase in the average number of filters per query across iterations. At the final iteration, DeepShop queries include, on average, 1.95 more filters than the seed queries, with the hard subset further increasing this difference to 2.88 filters on average. Similarly, the evolution of sorting preferences, illustrated in Figure 4(c), shows an upward trajectory. The final average sorting preferences per query exceed the seed data by 0.37, and this increment is further pronounced in the hard subset, where queries contain an additional 0.66 sorting preferences on average.

More detailed analysis of task complexity distribution across domains is provided in Appendix B.

### 3.6 Evaluation Metrics

To comprehensively evaluate web agents within the DeepShop environment, we adopt a two-stage evaluation protocol that includes both fine-grained evaluation and holistic task success evaluation.

**Fine-grained evaluation.** Given the cost and scalability challenges of human evaluation, following previous work (He et al., 2024a; Xue et al., 2025), we use GPT-4o for automatic evaluation. We first decompose each query into product attribute $q_{\text{attr}}$, search filter $q_{\text{filter}}$ and sorting preference $q_{\text{sort}}$ subqueries. For each web agent trajectory, we prompt GPT-4o to assess whether the final results align with the requirements specified in each subquery. Specifically, we prompt GPT-4o with the user subquery, screenshots, and the final answer of web agents and prompt GPT-4o to provide a binary decision ("Success" or "Not Success") for each subquery. This fine-grained evaluation enables us to capture partial success cases and diagnose failure modes more precisely than holistic binary task success alone. Note that if a particular subquery is not present in the original query (i.e., None), we skip the evaluation for that aspect and do not include it in the calculation.

**Holistic evaluation.** To calculate the overall task success, we rely on the above fine-grained evaluation outcomes, specifically the success scores for product attribute, search filter, and sorting preference. The holistic evaluation aggregates these components by rule-based checking, for each dimension, whether the query explicitly specifies a requirement. If a particular aspect (e.g., attribute, filter, or sorting) is present in the query, its corresponding success score is considered; otherwise, the system treats it as automatically satisfied. The final holistic task success is determined only if all required components are successfully satisfied: the system must meet all attribute, filter, and sorting requirements that are explicitly part of the query. For deep research systems, since intermediate execution screenshots are unavailable, fine-grained and holistic evaluations are conducted manually.

**Agreement rate between LLM evaluation and human judge.** Following previous work (He et al., 2024a; Xue et al., 2025), we calculate the agreement of human and GPT-4o judgments to measure the reliability of GPT-4o evaluations. Specifically, human annotators are shown the full interaction trace of the agent, including screenshots and actions, and are asked to judge whether the agent successfully fulfilled the user's request. Finally, the agreement rates between human and GPT-4o judges for the product attributes, search filters and sorting preferences, and overall task success are 84%, 80%, 82%, and 86%, respectively. It indicates the effectiveness and reliability of GPT-4o evaluation.

More details of the evaluation are provided in Appendix C.

## 4 EXPERIMENTS

### 4.1 RESEARCH QUESTIONS

We aim to answer the following research questions in our experiments: **RQ1**: How do simple RAG methods, web agents and deep research systems perform on the DeepShop benchmark in terms of fine-grained and holistic evaluation metrics? **RQ2**: How do existing methods perform across different product categories (Books, Electronics, Home, Fashion and Sports) in online shopping tasks? **RQ3**: How does the performance of web agents vary across different levels of query complexity, from seed queries to evolved complex queries with multiple attributes, filters and sorting preferences?

### 4.2 BASELINES

We evaluate web agents against three baseline categories:

- **Simple RAG**: Combines GPT-4o with Google Search by submitting the query, retrieving the top-ranked page, and generating a response based on webpage screenshots.
- **Web agents**: **Agent-E** (Abuelsaad et al., 2024) uses a hierarchical planner-actor framework with DOM tree distillation. **SeeAct** (Zheng et al., 2024) and **WebVoyager** (He et al., 2024a) use LLM multimodality, combining visual perception with action planning. **Browser Use** (Müller & Žunič, 2024) integrates visual understanding and HTML extraction for robust interaction.
- **Deep research systems**: **Gemini Deep Research** (Gemini, 2025) decomposes queries and generates cited multi-step reports using Gemini's extended reasoning. **OpenAI Deep Research** (OpenAI, 2025) autonomously browses, analyzes, and synthesizes web information into citation-rich outputs, emulating human research workflows.

More details on the baselines and implementation are provided in Appendix D and Appendix E.

## 5 EXPERIMENTAL RESULTS AND ANALYSIS

Table 2: Main results across product attributes, search filters, sorting preferences, and overall task success rates. Underlined indicates the best performance in web agents, and **bold** highlights the best performance in deep research systems.

| Method | Product attribute | Search filter | Sorting preference | Task success |
|---|---|---|---|---|
| *Simple RAG* | | | | |
| GPT-4o + Google Search | 7.33 | 5.97 | 4.55 | 7.33 |
| *Web agents* | | | | |
| Agent-E | 12.67 | 9.70 | 3.41 | 6.67 |
| SeeAct | 52.00 | 22.39 | 20.45 | 10.67 |
| WebVoyager | 40.67 | 38.00 | 23.86 | 16.00 |
| Browser Use | 36.00 | 34.33 | 30.68 | 32.00 |
| *Deep research systems* | | | | |
| Gemini Deep Research | 53.33 | 44.00 | 52.94 | **30.00** |
| OpenAI Deep Research | **60.00** | **46.15** | **58.82** | **30.00** |

### 5.1 PERFORMANCE ANALYSIS OF WEB AGENTS (RQ1)

We present the experimental results for the simple RAG baseline, web agents and deep research systems in Table 2. For each baseline, we evaluate the success rate of three fine-grained aspects, product attributes, search filters, and sorting preferences, as well as the holistic task success rate. Based on these results, we have three main observations:

- **Simple RAG methods fail to solve DeepShop due to the lack of website interaction capabilities.** We observe that simple RAG methods perform poorly across both fine-grained and holistic evaluations, with all success rates below 8%. In particular, these methods struggle with search filters (score: 5.97) and sorting preferences (score: 4.55), as such requirements cannot be satisfied through retrieval alone but instead demand active interaction with website elements (e.g., buttons).

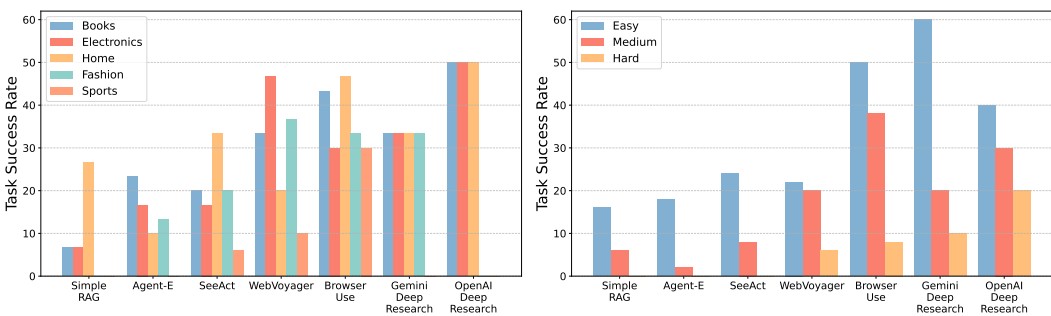

(a) Performance across different product categories.  (b) Performance across query complexity evolution.

Figure 5: Detailed analysis of performance across different product categories and query complexity.

This highlights the inherent complexity of DeepShop queries and underscores the need for agents capable of dynamic web interaction.

- **Web agents outperform simple RAG by using website interaction, but still struggle with DeepShop's fine-grained requirements.** Web agents dynamically interact with site content, enabling more effective product discovery than simple RAG. We observe progressive gains in overall task success: from HTML-based Agent-E (score: 6.67) to vision-based SeeAct (score: 10.67) and WebVoyager (score: 16.00), peaking with Browser Use (score: 32.00), which integrates HTML and visual inputs. Notably, SeeAct excels in product attributes, WebVoyager in search filters, and Browser Use in sorting preferences. However, satisfying all three dimensions simultaneously remains difficult, underscoring the challenge DeepShop poses for web agents.

- **Deep research systems use multi-step reasoning to enhance fine-grained performance on DeepShop, but overall success remains limited.** Gemini and OpenAI deep research systems excel in product attribute and sorting preference evaluations, outperforming web agents on these aspects. They struggle with search filters, as many require deep exploration and confirmation on product detail pages. Despite achieving higher fine-grained success, their holistic task success rates (30% each) remain low, underscoring the difficulty of satisfying all DeepShop requirements simultaneously and highlighting the benchmark's challenge for powerful deep research systems.

## 5.2 PERFORMANCE ACROSS DIFFERENT PRODUCT CATEGORIES (RQ2)

Figure 5(a) shows a performance analysis across product categories. **Agent performance varies notably across product categories.** The simple RAG method performs relatively well in *Home*, benefiting from rich textual titles retrievable via Google Search, but drops to 0% success in *Fashion* and *Sports*, where visual cues dominate. HTML-based Agent-E consistently underperforms, particularly in *Sports*, due to its inability to process visual content. Vision-based agents like SeeAct and WebVoyager improve performance across domains, while the hybrid Browser Use achieves the best cross-domain results by combining HTML and vision inputs. Deep research systems show relatively stable trends across categories but face major challenges in *Fashion* and *Sports*, where Gemini scores 0% in *Sports* and OpenAI fails entirely in both. These failures highlight the need for robust multimodal reasoning to handle visually driven product categories effectively.

## 5.3 PERFORMANCE ACROSS QUERY COMPLEXITY EVOLUTION (RQ3)

We analyze baseline performance across increasing query complexity, as shown in Figure 5(b). **Our results reveal a clear negative correlation between query complexity and agent performance.** Tasks are grouped into easy (0–1 complexity evolution), medium (2–3), and hard (4–5). The simple RAG method achieves success rates of 16% on easy and 6.00% on medium queries but drops to 0% on hard tasks, showing that Google Search alone cannot handle complex user needs. Web agents also exhibit sharp declines, with average accuracy falling from 28.5% (easy) to 17% (medium), then further dropping by 7 percentage points on hard tasks. Notably, deep research systems perform better than web agents on the hard subset, highlighting the importance of strong reasoning capabilities—yet even the top-performing OpenAI system reaches only 20% success rate.

## 5.4 ERROR ANALYSIS AND FUTURE IMPROVEMENT GUIDANCE

We conduct a detailed error analysis to identify the primary failure modes of web agents and deep research systems during task execution. Understanding these issues is critical for designing more robust and effective shopping agents. We categorize the observed errors as follows:

- **Web agents are limited by grounding ability.** HTML content and webpage screenshots provide complementary signals. Agents relying solely on HTML often overlook visual details—such as product color or layout cues—that are crucial for correct decisions. Conversely, vision-based agents using set-of-mark prompts struggle with segmentation accuracy: interactive buttons are frequently misclassified, and regions like customer reviews remain unsegmented, preventing the use of rating filters. Additionally, small filtering and sorting widgets are often ignored, degrading task performance. Future work may explore multimodal fusion techniques that combine HTML structure with visual context to enable stronger grounding (Gou et al., 2024).
- **Web agents often lack state assessment and replanning capabilities.** Agents frequently issue overly specific search queries and, upon retrieval failure, fail to backtrack or reformulate broader alternatives. Similarly, after navigating to product detail pages and finding unmet requirements, they rarely reconsider or explore other options. This lack of dynamic replanning leads to suboptimal decisions. Moreover, due to limited awareness of webpage state transitions, agents tend to repeat ineffective actions, such as clicking the same unresponsive element multiple times, instead of adjusting their strategy. Future research could fine-tune agents on realistic web environments to enhance their ability to reason over search failures, and adapt plans dynamically (Liu et al., 2025).
- **Web agents are constrained by a limited action space.** Web agents operate within a restricted set of browser actions, which prevents interaction with dynamic UI components found on shopping platforms. E.g., a web agent fails to filter products within a specific range because it cannot drag the price slider. More broadly, agents struggle to operate dropdowns, sliders, and nested menus—essential actions for precise filtering and sorting. Future work could expand the agent's action set with shopping-specific operations and deeper browser integration (Xue et al., 2025).
- **Web agents lack the ability to learn from execution.** Current agents show little ability to generalize across tasks. Experiences gained during one interaction—e.g., which strategies led to success or failure—are not transferred to future scenarios. Agents repeatedly make the same mistakes and fail to exploit previously effective strategies. Enabling execution-time learning and memory would allow agents to abstract successful patterns, track failure cases, and refine their decision-making over time. Future research may explore task-level memory, outcome-based self-reflection, and lifelong learning (Wang et al., 2025; Zheng et al., 2025).
- **Deep research systems are prone to hallucination errors.** OpenAI's deep research systems often oversimplify complex queries, neglecting constraints and returning confident yet inaccurate recommendations. For instance, they may assert that a matching product exists even when it does not. Although Gemini more frequently acknowledges failure and suggests approximate alternatives, both systems frequently return incomplete or incorrect links—redirecting to irrelevant websites or generic navigation pages rather than specific product detail views. These hallucinations reduce trust and usability. Future work could apply preference alignment and fact-checking techniques to reduce hallucination rates and improve the precision of retrieved links (Song et al., 2025).

More details of our error analysis are provided in Appendix F.

## 6 CONCLUSIONS

In this paper, we introduce DeepShop, a benchmark aimed at evaluating web agents in realistic and complex online shopping environments. While existing benchmarks often rely on simplistic and deterministic queries, DeepShop bridges this gap by incorporating real-world user intents and progressively evolving both the diversity and complexity of queries. Our benchmark covers five major e-commerce domains and evaluates agent performance across key dimensions, including product attributes, search filters, and sorting preferences. To enable a comprehensive assessment, we propose a fine-grained and holistic evaluation framework. Experimental results on recent web agents reveal significant performance drops on complex queries, highlighting the need for more robust agent design. Overall, DeepShop provides a challenging and realistic testbed for advancing the development of intelligent, user-centered web shopping agents.

## REPRODUCIBILITY STATEMENT

To ensure the reproducibility of our work, the source code for our evaluation scripts and dataset evolution, along with the DeepShop dataset, is available at `https://anonymous.4open.science/r/DeepShop-E4DF`. These resources allow researchers to replicate our experiments and verify the results reported in this paper.

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

APPENDIX

# A    USE OF LLMS

In this work, LLMs are used in two ways. First, they are used exclusively as a general-purpose tool for language refinement and manuscript polishing. Second, LLMs are employed as evaluators, and their usage in this role is described in detail in the paper. We conducted experiments to ensure the reliability of LLM-based evaluation. LLMs did not contribute to the conceptualization, experimental design, data analysis, or interpretation of the results in this work. All scientific content, findings, and conclusions presented in this paper are the sole responsibility of the authors. No text generated by LLMs affects the originality or intellectual contribution of the work.

# B    DETAILS OF TASK COMPLEXITY DISTRIBUTION ANALYSIS

To further validate that complexity evolution introduces consistent task difficulty across different domains in the DeepShop benchmark, we conduct a detailed analysis of overall task complexity distribution across domains.

For the overall task complexity distribution, we calculate the mean and standard deviation of the total number of product attributes, search filters, and sorting preferences. As shown in the table 3, DeepShop exhibits a higher average number of constraints and lower standard deviations across domains compared to the seed datasets. These results suggest that DeepShop presents greater overall task difficulty while maintaining improved cross-domain consistency.

Table 3: Overall task complexity distribution across domains.

| Domain | Electronic | Fashion | Sports | Books | Home | Avg. | Std |
|---|---|---|---|---|---|---|---|
| Seed data | 3.43 | 3.60 | 3.29 | 4.25 | 3.50 | 3.61 | 0.33 |
| DeepShop | 6.22 | 6.30 | 6.32 | 6.26 | 6.11 | 6.24 | 0.07 |

# C    DETAILS OF EVALUATION

## C.1    GPT-4O FOR EVALUATION

This section details the GPT-4o prompt used for fine-grained evaluation. As shown in Figure 6, we adapt the prompt from prior work (He et al., 2024a) to assess whether the web agent satisfies the requirements related to product attributes, search filters, and sorting preferences. For each trajectory, GPT-4o receives the subqueries, the agent's action history, and a maximum of 15 screenshots, and is prompted to determine whether the task was successfully completed.

## C.2    HUMAN EVALUATION

To validate the reliability of GPT-4o-based evaluation, we conduct a human evaluation on 50 randomly sampled trajectories from WebVoyager. Following previous work (He et al., 2024a; Xue et al., 2025), we hire human annotators with at least an undergraduate degree, and each sampled trajectory is independently evaluated by a minimum of two annotators. In cases of disagreement, a third annotator is consulted, and the final judgment is determined by majority vote. The evaluation is conducted based on the full execution trace of the agent, including the task description, operation history, and screenshots, as shown in Figure 7. Each human annotator labels the success of each subgoal—product attributes, search filters, and sorting preferences—as well as the overall task completion. The average Cohen's kappa among human annotators is 0.82, indicating substantial inter-annotator agreement. We then compute the agreement rate between human and GPT-4o assessments.

Additionally, since deep research systems do not expose intermediate screenshots or action histories, human annotators are allowed to access the Amazon website to verify whether the returned links satisfy the specified requirements. Due to limited use of the deep research systems, we evaluated its performance on 30 randomly sampled queries. To ensure consistency, we evaluate the accuracy based on the first recommended item when multiple products are returned.

[System prompt]
As an evaluator, you will be presented with three primary components to assist you in your role:

1. Web Task Instruction: A clear and precise natural language directive that specifies an online shopping activity to be executed. The instruction may involve locating products that meet certain attribute requirements (e.g., color, size, brand), applying specific search filters (e.g., price range, customer ratings, availability), or fulfilling user-defined sorting preferences (e.g., lowest price, newest arrivals, best sellers). Tasks may also include verifying product details, comparing offers, or checking for shipping and return policies, depending on the scenario.

2. Result Screenshots: This is a visual representation of the screen showing the result or intermediate state of performing a web task. It serves as visual proof of the actions taken in response to the instruction.

3. Result Response: This is a textual response obtained after the execution of the web task. It serves as textual result in response to the instruction.

-- You DO NOT NEED to interact with web pages or perform actions such as conducting searches on websites.
-- You SHOULD NOT make assumptions based on information not presented in the screenshot when comparing it to the instructions.
-- Your primary responsibility is to conduct a thorough assessment of the web task instruction against the outcome depicted in the screenshot and in the response, evaluating whether the actions taken align with the given instructions.
-- NOTE that the instruction may involve more than one task, for example, locating the garage and summarizing the review. Failing to complete either task, such as not providing a summary, should be considered unsuccessful.
-- NOTE that the screenshot is authentic, but the response provided by LLM is generated at the end of web browsing, and there may be discrepancies between the text and the screenshots.
-- Note the difference: 1) Result response may contradict the screenshot, then the content of the screenshot prevails, 2) The content in the Result response is not mentioned on the screenshot, choose to believe the content.

You should elaborate on how you arrived at your final evaluation and then provide a definitive verdict on whether the task has been successfully accomplished, either as 'SUCCESS' or 'NOT SUCCESS'.

[User prompt]
TASK:
{subquery}

Result Response:
{answer}

15 screenshots at the end:
{screenshots}

Figure 6: Prompts for GPT-4o evaluation

> You will be presented with a web shopping task.
> For each task, you will receive three subqueries, along with the web agent's action history and corresponding screenshots. Your goal is to evaluate the agent's performance across three specific dimensions: product attributes, search filters, and sorting preferences. Please note: if a subquery is labeled as None, you do not need to assess that particular aspect.
>
> Definitions of the three subqueries are as follows:
> 1. Product attributes, referring to concrete product characteristics users may specify to express detailed intent
> 2. Search filters, representing categorical or numerical constraints commonly used on e-commerce platforms
> 3. Sorting preferences, indicating desired result orderings, such as price or popularity.
>
> Task:
> {Query}
>
> Product Attribute Requirement:
> {Subquery1}
>
> Search Filter Requirement:
> {Subquery2}
>
> Sorting Preference Requirement:
> {Subquery3}
>
> Agent Action History:
> {Action}
>
> Screenshots:
> {Screenshots}

Figure 7: Instructions for human evaluation.

### C.3 SIGNIFICANCE TEST BETWEEN WEB AGENTS

We conducted paired t-tests to compare the overall task success rates between **Browser Use** and four baseline models: **Simple RAG**, **Agent-E**, **SeeAct**, and **WebVoyager**. The results reveal that **Browser Use significantly outperforms Simple RAG and web agent baselines** ($p < 0.05$), supporting the claim that direct interaction with the online web environment can substantially improve agent performance over static retrieval methods such as Simple RAG. Furthermore, the comparison highlights fundamental limitations in grounding capabilities across different agent types. HTML-based agents like Agent-E lack access to visual context, often missing essential cues such as product color or spatial layout. On the other hand, vision-based agents such as SeeAct and WebVoyager rely on set-of-mark prompting, which suffers from segmentation errors: interactive elements are frequently misclassified, and key sections—like customer review areas—remain unsegmented, making it difficult to apply filters like rating thresholds.

## D DETAILS OF BASELINES

- **Simple RAG method**: We implement a simple retrieval-augmented generation (RAG) baseline by combining GPT-4o with Google Search. The user query is first submitted to Google Search, and the top-ranked webpage is selected. GPT-4o (version 2024-08-06) then generates a final response conditioned on the screenshot of the retrieved page. We use the Serper API[1] to programmatically access Google search results.
- **Web agents**: For fair comparison, all web agents are instantiated using GPT-4o (version 2024-08-06) as the underlying language model. **Agent-E** (Abuelsaad et al., 2024) is an HTML-based agent that adopts a hierarchical planner-actor architecture, augmented with flexible DOM tree distillation and a denoising mechanism to improve decision accuracy.[2] **SeeAct** (Zheng et al., 2024) exploits the multi-modal capabilities of large language models (LLMs), integrating visual perception with

---

[1] https://serper.dev/
[2] https://github.com/EmergenceAI/Agent-E

structured web-based interactions.[3] **WebVoyager** (He et al., 2024a) also leverages multi-modal reasoning and introduces a set-of-mark prompting scheme that guides the agent to first generate intermediate thoughts before selecting final actions.[4] **Browser Use** (Müller & Žunič, 2024) is an open-source web agent framework that combines visual understanding with HTML structure parsing to support robust web navigation and interaction.[5]

• **Deep research systems**: Since we cannot strictly constrain deep research systems to operate on specific shopping websites, we include explicit site constraints in the prompt to guide the search process. The prompt format used for these systems is illustrated in Figure 8. **Gemini Deep Research** (Gemini, 2025) is an AI assistant integrated into Google's Gemini Advanced platform. We evaluate the Gemini 2.0 Flash model with deep research capabilities.[6] **OpenAI Deep Research** (OpenAI, 2025) is an agentic system powered by OpenAI's reasoning models. We evaluate the o3 model with deep research enabled.[7]

---

Prompt:
Your task is to help find relevant products on Amazon website
(https://www.amazon.com/?language=en_US&currency=USD) based on the following question:
{Query}

Please strictly follow the rules:
1. Do not contact the user for further information. Provide answers directly without asking any questions or seeking clarification.
2. Include Amazon product links for all recommended items.

---

Figure 8: Prompts for deep research systems.

# E DETAILS OF IMPLEMENTATION

## E.1 PROMPTS FOR QUERY DIVERSITY AND COMPLEXITY EVOLUTION

Details for the prompts used in $\text{Diversity}(\cdot)$, and $\text{Complexity}(\cdot)$ are provided. Figures 9, 10, and 11 display the prompts for query diversity, complexity evolution and detailed complexity evolution strategy, respectively.

## E.2 IMPLEMENTATION DETAILS

We evaluate open-source agents—Agent-E, SeeAct, WebVoyager, and Browser Use—within real-time web environments. Agent-E, SeeAct, and Browser Use are executed via Playwright, while WebVoyager leverages Selenium. To control computation cost and prevent excessive exploration, we limit each agent to a maximum of 15 steps per task. All agents are powered by `gpt-4o-2024-08-06` as the underlying language model. For automatic evaluation, we also adopt `gpt-4o-2024-08-06` as evaluators, with temperature set to 0 to reduce response variance and enhance reproducibility. The agents differ in their perception mechanisms: Agent-E and SeeAct utilize full-page screenshots, whereas WebVoyager and Browser Use operate on the visible viewport only.

# F DETAILS OF ERROR ANALYSIS

• **Web agents are limited by grounding ability.** As shown in Figure 12, agents struggle to accurately ground interface elements. For example, button 39 related to user rating was not properly segmented, preventing the agent from selecting a specific rating range. Buttons 31–37 and 41–44 were rendered too densely and overlapped, making interaction difficult. Additionally, the sorting button on the right was incorrectly split into two buttons 16 and 17, which may confuse the agent during execution.

---

[3] https://github.com/OSU-NLP-Group/SeeAct
[4] https://github.com/MinorJerry/WebVoyager
[5] https://github.com/browser-use/browser-use
[6] https://blog.google/products/gemini/google-gemini-deep-research/
[7] https://openai.com/index/introducing-deep-research/

I want you act as a Prompt Creator.
Your goal is to draw inspiration from the #Given Prompt# to create a brand new prompt.
This new prompt should be in the web shopping domain but tailored for different specific products in the {Randomly selected product category} amazon product field.
The LENGTH and complexity of the #Created Prompt# should be similar to that of the #Given Prompt#.
The #Created Prompt# must be reasonable and must be understood and responded by humans.
'#Given Prompt#', '#Created Prompt#', 'given prompt' and 'created prompt' are not allowed to appear in #Created Prompt#

#Given Prompt#:
{Original query}

#Created Prompt#:

Figure 9: Prompts for query diversity evolution.

I want you act as a Prompt Rewriter for web shopping.
Your objective is to rewrite a given prompt into a more complex version to make those web shopping agents a bit harder to handle.
But the rewritten prompt must be reasonable and must be understood and responded by humans.
You should complicate the given prompt using the following method:
{Randomly selected evolution strategy}
You should try your best not to make the #Rewritten Prompt# become verbose, #Rewritten Prompt# can only add 10 to 20 words into #The Given Prompt#.
'#The Given Prompt#', '#Rewritten Prompt#', 'given prompt' and 'rewritten prompt' are not allowed to appear in #Rewritten Prompt#

#Given Prompt#:
{Original query}

#Created Prompt#:

Figure 10: Prompts for query complexity evolution.

**Product attribute evolution:**
Enhance #The Given Prompt# by integrating detailed product attributes that detail user needs. Please specifies concrete values for one product attribute (e.g., brand, model, price range, color, size, weight, or unique features) based on your knowledge about this product, ensure that these exact details are incorporated into the query instead of using generic placeholder terms.

**Search filter evolution:**
Enhance #The Given Prompt# by integrating detailed product constraints that capture user needs. Please specifies concrete values for constraints—such as a minimum customer rating (e.g., above 4.0 or 4.5 stars), a minimum number of customer reviews (e.g., 100, 300, 500, or 1000), shipping options like free delivery, new arrival time frames (e.g., released in the last 30 or 90 days), return policies (e.g., free returns), or warranty information (e.g., includes a 1-year warranty) based on your knowledge about amazon website, ensure that these exact values are used in the query rather than generic terms.

**Sorting preference evolution:**
Enhance #The Given Prompt# by integrating a specific product filtering requirement for web shopping. Find the top product based on one of the following criteria: lowest price, highest user rating, newest arrival, or best seller ranking.

Figure 11: Prompts for complexity evolution strategies.

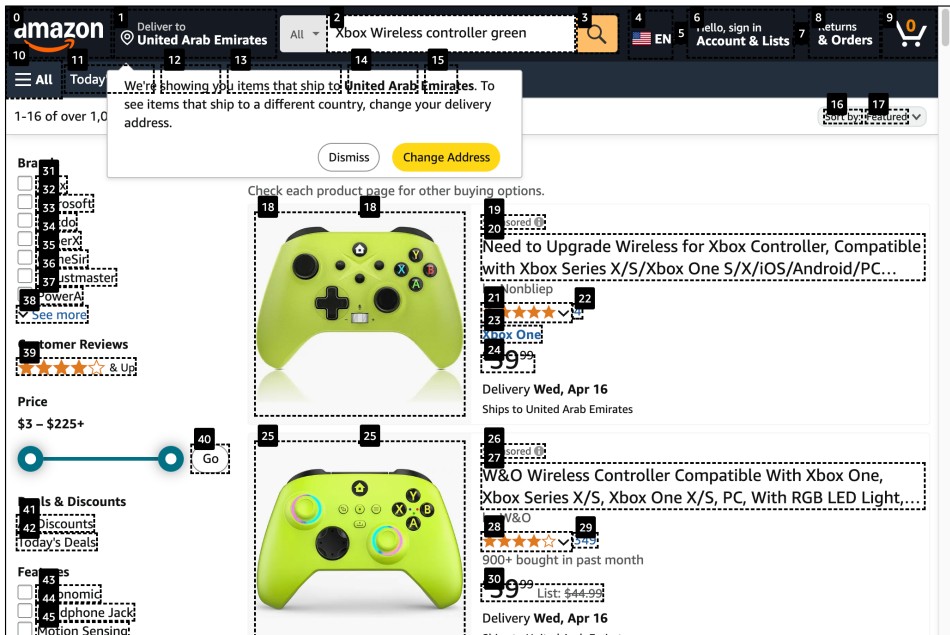

Figure 12: Limited grounding ability of web agents.

- **Web agents often lack state assessment and replanning capabilities.** As illustrated in Figure 13, when the agent enters a product detail page to verify a 1-year warranty, it fails to reassess its state upon realizing that the requirement is unmet. Instead of returning to the search results page, the agent continues to scroll within the current page, inefficiently attempting to locate an alternative product.

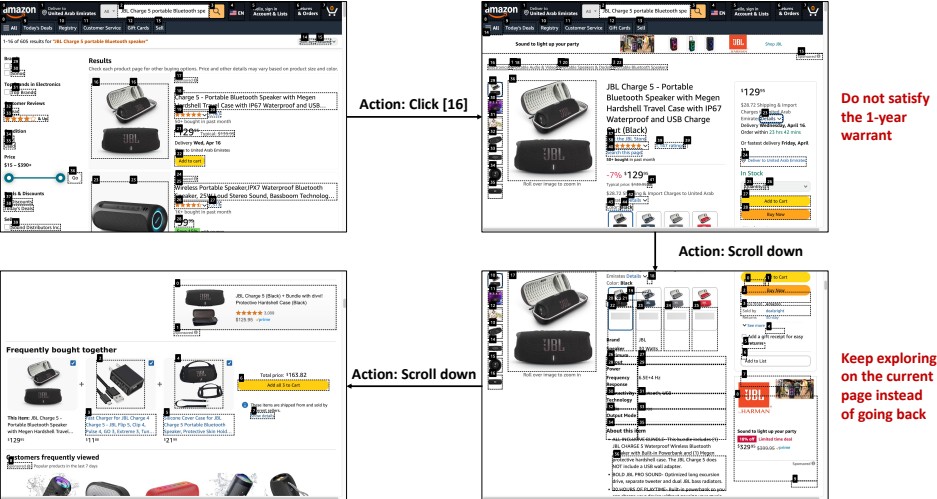

Figure 13: Illustration of web agent's failure to reassess and replan.

- **Web agents are limited by action space.** As shown in Figure 14, the agent attempts to filter cameras within the $100–$300 price range but fails to do so due to its inability to interact with dynamic UI elements such as price sliders. Instead, it clicks the adjacent "Go" button without adjusting the slider values, resulting in ineffective filtering. This highlights a fundamental limitation of current agents: the constrained action space prevents them from performing fine-grained interactions required in realistic web environments.
- **Web agents lack the ability to learn from execution.** As illustrated in Figure 15, we present screenshots of the web agent across four different tasks. A consistent failure pattern emerges: the agent repeatedly misuses the retriever to query filtering or sorting constraints, despite these functionalities being accessible only via the designated filter or sorter UI components. Due to the retriever's inability to interpret such structural intents, it often returns irrelevant results

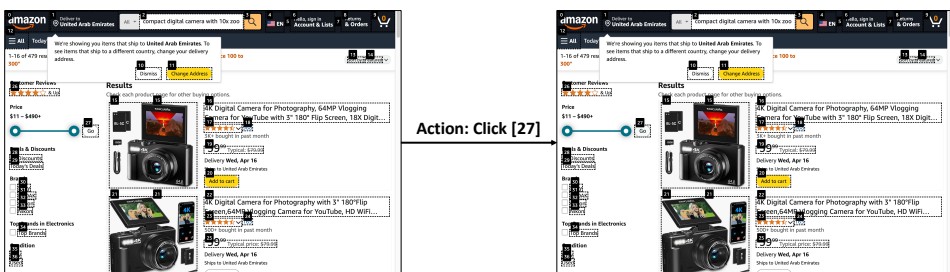

Clicked the 'Go' button in the price block but failed to filter
prices to the $100–$300 range.

Figure 14: Illustration of the web agent's failure to apply the price filter during task execution.

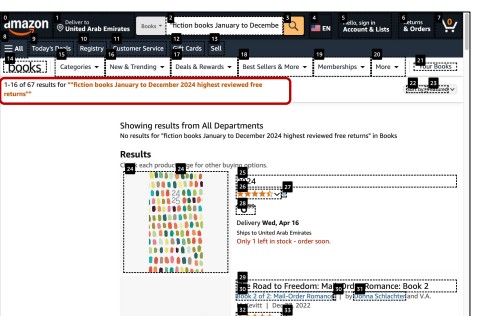
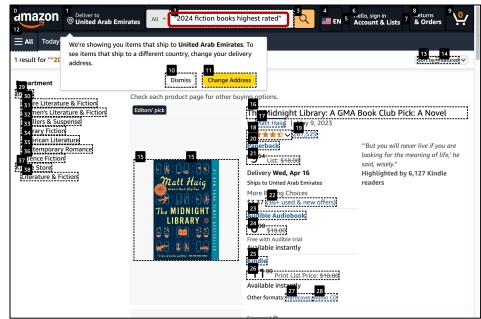

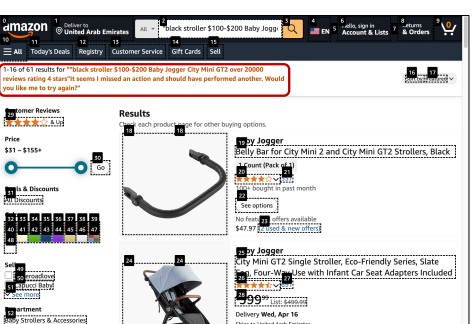
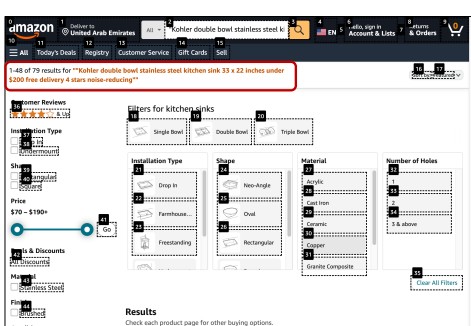

Figure 15: Illustration of the web agent's failure to learn from execution.

contaminated by noise. This repetition across tasks highlights the agent's lack of execution-time learning, preventing it from abstracting past mistakes and adapting its behavior accordingly.

- **Deep research systems are limited by hallucination errors.** As shown in Figure 16, the task requires *identifying a Women's Vintage Floral Maxi Dress in Navy Blue, Size: Medium, and explaining the return policy if free returns are available*. However, as shown in Figure 17, the first link returned by the OpenAI deep research system points to a product only available in Large and XX-Large sizes. Despite this mismatch, the system hallucinates that the size requirement is met. Moreover, Link2 and Link3 direct to non-Amazon websites, which violates the task constraint. In an attempt to explain the return policy, the system incorrectly extracts return information from these external sites. These issues reveal that when facing complex and fine-grained shopping queries, the OpenAI deep research system frequently exhibits hallucinations—both in satisfying attribute constraints and in sourcing policy information from inappropriate domains.

---

**Task:**
Locate the Return Policy for Women's Vintage Floral Maxi Dress on Amazon. Color: Navy Blue, Size: Medium. If free returns are available, please explain the return process for this item.

**OPENAI deep research system answer:**
Understood. I will search Amazon for listings of a Women's Vintage Floral Maxi Dress in Navy Blue, size Medium, and identify the return policy for each listing. I will focus on whether free returns are available and explain the return process for the item. I'll share the direct Amazon links for all relevant listings I find.

Research completed in 11m:

Return Policy for Women's Vintage Floral Maxi Dress on Amazon
**Free Return Policies for Navy Blue Vintage Floral Maxi Dresses (Medium)**
When shopping on Amazon for a women's vintage floral maxi dress in Navy Blue (size Medium), it's important to check if the item offers Free Returns. Below are a few popular listings that match this description, along with their return policy details:
**Floerns Women's Sleeveless Halter Neck Vintage Floral Print Maxi Dress (Navy Blue)**
**Product:** *Floerns Women's Sleeveless Halter Neck Vintage Floral Print Maxi Dress*
**Color:** "A Navy Flower",
**Size:** Medium.
**Link:** Amazon Product Page – Floerns Halter Neck Floral Maxi
**Free Returns: Yes.** This Floerns dress is Prime-eligible and marked with "FREE Returns", meaning you can return it at no cost if it doesn't fit or meet your expectations [Link1]. Amazon provides at least one prepaid return shipping option for eligible items, so returning this dress is hassle-free.
**Return Process:** You have 30 days from delivery to initiate a return via your Amazon account. Simply go to Your Orders, select this dress, and choose "Return or Replace Items."
Because it's eligible for free returns, Amazon will offer a prepaid shipping label or QR code for drop-off [Link2]. You can then package the dress (in new, unworn condition with tags) and drop it off at an approved location (e.g. UPS Store, Whole Foods, Kohl's, etc.) without paying return shipping [Link3]. Once the item is received, you'll get a full refund to your original payment method.

[Link1]: https://www.amazon.com/Floerns-Womens-Sleeveless-Halter-Vintage/dp/B0C3HQLBBQ#:~:text=,least%20one%20free%20return
[Link2]: https://www.smoking-er.com/store/rosemary-pellet#:~:text=,preferred%20free%20shipping%20option%20Drop%C2%A0off%C2%A0and%C2%A0leave
[Link3]: https://www.smoking-er.com/store/rosemary-pellet#:~:text=,preferred%20free%20shipping%20option%20Drop%C2%A0off%C2%A0and%C2%A0leave

---

Figure 16: Illustration of hallucination errors in the OpenAI deep research system. The system returns three links for the query, including two that point to non-Amazon websites, and incorrectly assumes the returned products match the specified size and platform constraints.

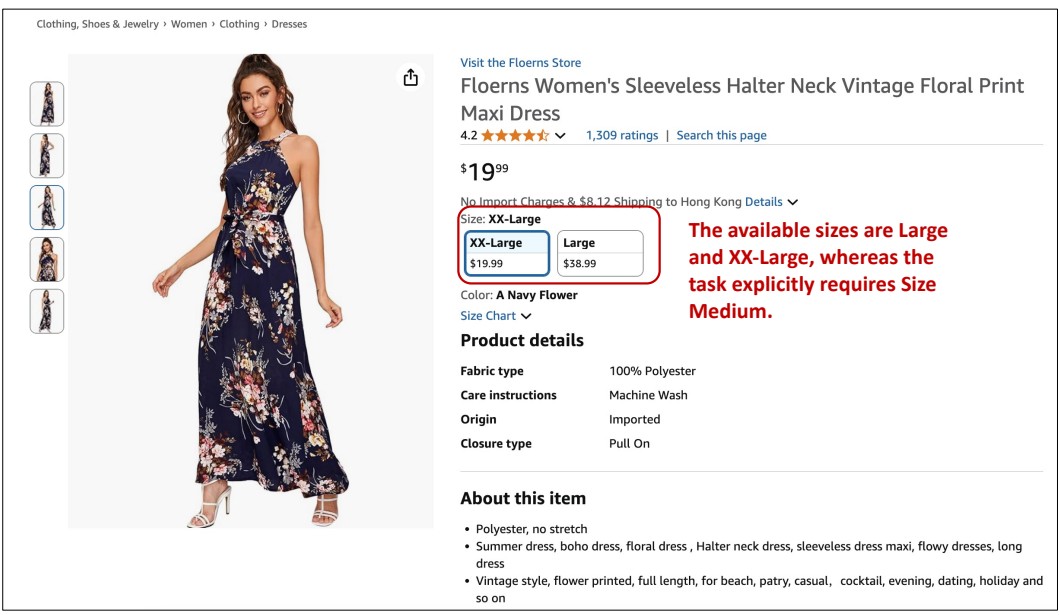

Figure 17: Detailed view of the first returned product link. Although the task specifies size Medium, the linked product only offers Large and XX-Large options.

