# OpenReview forum: "DeepShop: A Benchmark for Deep Research Shopping Agents"
_ICLR.cc/2026/Conference — Submitted to ICLR 2026_

### Official Review · Reviewer_C7Vr · 2025-10-30

**Soundness:** 2
**Presentation:** 3
**Contribution:** 1
**Rating:** 4
**Confidence:** 4

**Summary:**

This paper expands on existing online shopping benchmarks by increasing query complexity through the inclusion of product attributes, search filters, and sorting preferences. They demonstrate that increasing query complexity leads to decreased model performance, and that poor multimodal capabilities significantly harm performance.

**Strengths:**

- Increased complexity over existing benchmarks
- Identification of poor visual understanding as a bottleneck in some domains

**Weaknesses:**

- Lack of scope: while shopping is a relevant subdomain for web agents, this benchmark covers only a small slice of potential interactions, limiting potential impact.
- Lack of novelty: artificially increasing query complexity relative to existing benchmarks is only a minor variation on prior work, and does not introduce fundamentally new challenges.
- Limited insight: the fact that increasing the complexity of queries leads to worse performance is unsurprising.

**Questions:**

- How do the authors see this benchmark being used to guide further model or algorithm development, and how does it improve on this vs existing benchmarks?
- Does the increase in complexity vs WebShop elicit novel insights or behaviours not already visible in this benchmark?

**Details Of Ethics Concerns:**

- How do the authors see this benchmark being used to guide future algorithm development, vs existing web agent benchmarks?
- Does the increase in complexity vs WebShop reveal any novel insights or behaviours not already visible in these benchmarks?

---

> ### Author Response · Authors · 2025-11-23
> **Response to Reviewer C7Vr**
>
> We thank the reviewer for their time and feedback, which raises fundamental questions about the novelty and insight of our work. We appreciate this opportunity to clarify the unique contributions of DeepShop.
>
> **1. Regarding Novelty and Insight: A Shift from State Matching to Multi-step Reasoning**
>
> We agree that the observation "harder tasks are harder" is, on its own, unsurprising. However, we argue that the complexity introduced in DeepShop represents a **qualitative and fundamental shift** in the challenges posed to web agents, revealing novel failure modes that prior benchmarks could not.
>
> This qualitative difference is detailed in our benchmark comparison (Table 1). Prior benchmarks like WebShop [1] primarily test an agent's ability for **state matching**—finding a final product that satisfies a set of static attributes. In contrast, DeepShop is the first to systematically introduce and evaluate complex constraints like **search filters and sorting preferences**. This elevates the core challenge to **multi-step reasoning and execution**: agents must compile a complex instruction into an ordered sequence of operations (e.g., search -> apply rating filter -> sort results) and then reliably execute this "program" on a dynamic web interface.
>
> This qualitative shift reveals entirely new insights that are critical for guiding future research:
> *   **It exposes high-level planning failures, not just grounding failures.** Our analysis shows that agents often fail not because they cannot find a button, but because they fail in their high-level reasoning. For instance, an agent might correctly apply a filter but then *forget* the subsequent sorting instruction. This reveals critical limitations in the agent's planning and state-tracking capabilities, a failure mode that simpler, state-matching benchmarks cannot surface.
> *   **It demonstrates the brittleness of sequential actions.** We find that even agents that can perform single operations struggle to chain them reliably. Executing one step (e.g., filtering) changes the webpage's state, requiring the agent to re-evaluate and act robustly in this new context. DeepShop is the first benchmark to systematically test this crucial capability.
>
> Therefore, the key insight from DeepShop is not merely that complex tasks are hard, but that current state-of-the-art agents lack the fundamental **multi-step reasoning** required for real-world web automation. This provides a clear directive for future work: the community must focus not just on improving visual grounding, but on developing more sophisticated planners, state trackers, and architectures capable of executing complex, language-specified programs.
>
> **2. Regarding Lack of Scope**
>
> We acknowledge our focus on the e-commerce domain. This was a deliberate strategic choice to maximize the impact and generality of our findings.
>
> We chose e-commerce because it serves as a **microcosm of the most complex, dynamic, and interactive challenges** found on the modern web, a direction aligned with prior work that identifies the need to evaluate agents in realistic settings [1]. It is a domain that aggregates a dense variety of UI elements (sliders, nested menus, dynamic content) and user workflows. We posit that an agent capable of mastering the multi-step reasoning and robust interaction required by DeepShop will possess a core skill set that is highly transferable to other structured web domains, such as travel booking, academic search, or financial platforms. By stress-testing agents in this complex environment, we believe our findings offer the most valuable and generalizable insights for the broader web agent community.
>
> [1] Yao, S., et al. "WebShop: Towards Scalable Real-World Web Interaction with Grounded Language Agents."

---

> ### Comment · Area_Chair_jC2p · 2025-11-25
> **Please participate in discussions with authors and other reviewers asap**
>
> Please ensure you are actively participating in the discussion phase.
>
> Additionally, I strongly encourage you to read the other reviews and discuss with your fellow reviewers. It is vital to compare perspectives and raise any remaining concerns now to give the authors a fair opportunity to respond.
>
> Based on these interactions, please update your reviews and finalize your decisions.
>
> Best, AC

---

### Official Review · Reviewer_gWjR · 2025-10-31

**Soundness:** 3
**Presentation:** 3
**Contribution:** 3
**Rating:** 6
**Confidence:** 2

**Summary:**

This paper introduces DeepShop, a new benchmark designed to evaluate web agents and deep-research systems on realistic online shopping tasks that involve multi-faceted constraints—product attributes, filters, and sorting—over live e-commerce websites. The authors argue that prior web-agent benchmarks (e.g., Mind2Web-Live, WebVoyager) are either static or insufficiently complex, failing to capture the nuanced reasoning and interaction patterns needed for actual online shopping.

contributes:

A query evolution framework that expands and complexifies real shopping queries across five categories (Books, Electronics, Home, Fashion, Sports).

A multi-dimensional evaluation protocol that decomposes success into attributes, filters, sorting, and holistic task success.

Comprehensive empirical evaluation of both web agents and deep-research systems (e.g., GPT-4o + Browser Use, Gemini Deep Research, OpenAI Deep Research).

The findings reveal that current systems perform poorly on realistic multi-constraint queries—particularly on filter-related operations—highlighting major gaps between language reasoning and grounded web interaction.

**Strengths:**

Novel and practical benchmark: DeepShop fills an important gap by testing systems on real-time, constraint-heavy shopping tasks that mirror real user behavior.

Systematic query evolution: The staged expansion from simple to complex queries is methodologically sound and produces controllable difficulty tiers.

Granular evaluation: Separating fine-grained metrics (attributes, filters, sorting) from holistic success provides diagnostic insights that go beyond aggregate scores.

Comprehensive coverage: The benchmark spans five diverse categories and includes both web agents and deep-research systems, yielding generalizable findings.

Empirical rigor: The authors report inter-rater reliability between human and GPT-4o evaluators, adding credibility to automated scoring.

**Weaknesses:**

Limited task realism in some cases: Although queries evolve in complexity, the benchmark still relies on synthetic LLM-generated text rather than actual user intent logs, which may limit ecological validity.

Evaluation dependency on GPT-4o: Automated judging introduces potential bias, especially since GPT-4o is also part of the evaluated systems.

Unclear generalization beyond e-commerce: The authors claim DeepShop can inspire broader web-agent research, but no experiments are shown outside shopping domains.

Limited ablation or analysis of failure modes: The paper reports average scores but lacks detailed qualitative examples explaining why agents fail at filters or sorting.

**Questions:**

Human data grounding: How do you ensure that the evolved queries reflect realistic shopping intent? Did you validate them with human annotators or external user logs?

Judging consistency: Since GPT-4o is both a baseline and the automated evaluator, did you perform cross-model validation (e.g., using Gemini or Claude) to verify scoring robustness?

Dynamic web changes: Given that e-commerce websites change frequently, how do you ensure benchmark reproducibility over time?

Generalization potential: Could DeepShop’s query evolution and scoring framework be adapted for non-shopping domains such as travel booking or information retrieval?

Multi-modal grounding: Since categories like Fashion rely heavily on visual cues, how does DeepShop handle or evaluate the visual reasoning aspect?

---

> ### Author Response · Authors · 2025-11-23
> **Response to Reviewer gWjR**
>
> We thank the reviewer for the thoughtful and constructive review. We are encouraged that the reviewer found our benchmark novel and practical, and we appreciate the opportunity to clarify these important points.
>
> **1. Regarding Query Realism and Human Data Grounding**
>
> Thank you for this insightful question on ecological validity. We would like to clarify that our benchmark construction follows a **"LLM-generation + Human-review"** closed-loop process.
>
> While the query evolution is driven by an LLM to ensure systematic and scalable complexity, every query that was included in the final benchmark underwent a **rigorous manual review and calibration by human annotators**. This critical step ensures that each query is not only grammatically correct but also logically sound, plausible, and reflects realistic shopping intent. The LLM acts as a powerful candidate generator, but the final quality and realism are guaranteed by human oversight. We will add a detailed description of this human calibration process to the manuscript.
>
> **2. Regarding Evaluation Dependency on GPT-4o**
>
> This is a very important point regarding potential evaluation bias. We addressed this by grounding our automated evaluation in a **human-centric gold standard**.
>
> As reported in the main paper (Section 3.6), we have already established a **high agreement rate (80-86%) between GPT-4o's judgments and those of human evaluators**. Since human evaluation is the ultimate gold standard for correctness, this high level of inter-rater reliability strongly indicates that GPT-4o serves as a valid and unbiased proxy for human judgment in our setting. This validation step was crucial for us to ensure that the automated scoring is robust and not simply reflecting the biases of a single model. We will re-emphasize this point in the evaluation section to make the robustness of our protocol clearer.
>
> **3. Regarding Dynamic Web Changes and Reproducibility**
>
> We thank the reviewer for raising this key challenge for all online benchmarks. Our framework is designed for reproducibility by focusing on **"relational invariance" and "procedural correctness."** We evaluate whether an agent correctly executes an *action* (e.g., applying a "sort by price" filter) rather than whether it finds a product with a specific *value* that is subject to change. This ensures we are testing the agent's interaction capabilities. Furthermore, we will periodically verify the "solvability" of our tasks to maintain the benchmark's long-term utility.
>
> **4. Regarding Generalization Potential Beyond E-commerce**
>
> We appreciate the reviewer's prompt to think about broader applications. While we did not conduct experiments outside of e-commerce, our framework is designed to be conceptually generalizable.
>
> The framework consists of three modular components: (1) a set of seed queries, (2) a domain-specific "evolution grammar" (e.g., attributes, filters, sorting), and (3) a fine-grained evaluation protocol. This structure is highly adaptable. For instance, in a **"travel booking" domain**, the evolution grammar could be:
> *   **Attributes**: "non-stop flight," "business class."
> *   **Filters**: "free cancellation," "departure after 6 PM."
> *   **Sorting**: "shortest duration," "earliest arrival."
>
> This demonstrates that the core methodology for creating and evaluating complex, multi-constraint tasks can be readily ported to other information-seeking domains. We will add this example and discussion to the conclusion to better illustrate the generalization potential.
>
> **5. Regarding Analysis of Failure Modes and Visual Reasoning**
>
> Thank you for this excellent suggestion to deepen our analysis.
>
> Regarding failure modes, we would like to clarify that a detailed qualitative analysis, complete with illustrated case studies for each error category, is already provided in **Appendix F** of our paper. We apologize if this was not sufficiently highlighted in the main text. For instance, in Appendix F (Figure 12), we show a concrete example of a grounding failure on a filter button, and in Appendix F (Figure 14), we illustrate an agent failing to operate a price slider. We believe these detailed examples provide the diagnostic insights the reviewer is looking for.
>
> Regarding visual reasoning, we thank the reviewer for this important question. The primary goal of DeepShop is to evaluate an agent's ability to handle complex, multi-constraint shopping tasks. While we do not have a metric designed specifically to isolate visual reasoning, our results provide clear evidence of its importance. The significant performance gap between multimodal agents and text-only agents in visually-driven categories like Fashion and Sports (Section 5.2, Figure 5a) demonstrates that DeepShop effectively creates scenarios where visual grounding is critical for success. We agree that developing more explicit metrics for visual reasoning is a valuable direction, and we will include this as a key area for future work.

---

> ### Comment · Area_Chair_jC2p · 2025-11-25
> **Please participate in discussions with authors and other reviewers asap**
>
> Please ensure you are actively participating in the discussion phase.
>
> Additionally, I strongly encourage you to read the other reviews and discuss with your fellow reviewers. It is vital to compare perspectives and raise any remaining concerns now to give the authors a fair opportunity to respond.
>
> Based on these interactions, please update your reviews and finalize your decisions.
>
> Best, AC

---

### Official Review · Reviewer_znE7 · 2025-11-01

**Soundness:** 2
**Presentation:** 3
**Contribution:** 1
**Rating:** 2
**Confidence:** 3

**Summary:**

This paper proposes an in-the-wild evaluation suite for web navigation agents, specifically in the domain of shopping. The core contribution is a set of fine-grained instructions, e.g., “Find a queen-sized bedspread featuring a floral design in a calming blue hue…” along with automatic evaluators for determining whether a model has successfully executed these instructions.

**Strengths:**

1. Overall, the paper is pretty easy to read and understand

2. As far as I can tell, the task has a reasonable amount of headroom (cf. Table 2), suggesting that this benchmark might stay relevant for longer than more synthetic alternatives like WebShop

**Weaknesses:**

1. The paper may be of limited interest to researchers who are not specifically interested in web shopping agents. Much of its analysis is about, e.g., differences in agent performance across different shopping domains (e.g., books vs. fashion), and doesn’t seem applicable to web navigation or agent research more broadly. My impression is that the tasks described here are roughly a subset of those described in more general-purpose benchmarks like AssistantBench

2. One major concern I have is whether the tasks as specified are all achievable. Because the task design primarily involves intersecting a number of attributes (e.g., rating, # reviews), it seems possible that many of the generated user queries might not be uniquely resolvable. As a result, ceiling performance might be substantially lower than 100%

**Questions:**

1. How do human and LLM evaluators evaluate sorting preferences? It seems like this should condition not just on the final result but also on the full range of available options on the web.

2. Are models only allowed to search for products on Amazon? This is my impression from the appendix, but it should be clarified in the main body of the paper.

---

> ### Author Response · Authors · 2025-11-23
> **Response to Reviewer znE7**
>
> We are grateful to the reviewer for their valuable time and constructive feedback. We address the concerns raised below.
>
> **1. Regarding the Generality of Contribution**
>
> Thank you for this comment. We would like to clarify our perspective on the contribution's generality.
>
> We position online shopping not as a narrow application, but as a **microcosm for complex and general-purpose web interaction**. E-commerce websites integrate some of the most challenging UI elements for agents to master, including dynamic content, complex forms (filters, sliders, menus), and multi-page workflows. The skills required to succeed in DeepShop—such as deep navigation, fine-grained UI element grounding, and multi-step reasoning—are directly transferable to other complex web navigation domains (e.g., travel booking, academic search).
>
> While general-purpose benchmarks like AssistantBench cover a wide breadth of tasks, DeepShop provides a crucial test of **interaction depth and robustness** in a complex, real-world environment, which we believe is a valuable and complementary contribution to the broader agent research community. We will revise the introduction to better articulate this positioning.
>
> **2. Regarding Task Achievability**
>
> This is an excellent point, and we appreciate the opportunity to clarify our methodology.
>
> To address the concern of task solvability, every query in the final DeepShop benchmark underwent a **rigorous human-in-the-loop verification process**. As we briefly mention in Section 3.5, our research team manually executed each task on the target website at the time of benchmark construction to filter out any queries that could not be resolved. This ensures that the performance ceiling is indeed 100%. We acknowledge this crucial point was not prominent enough and will expand upon this verification process in the revised dataset creation section to make it clearer.
>
> **3. Regarding the Evaluation of Sorting Preferences**
>
> Thank you for this question about the evaluation protocol. We apologize for the lack of clarity.
>
> Our evaluation of sorting preferences relies on a **full trajectory review**. The evaluators (both human and LLM) are provided with the complete interaction trace of the agent, which includes a series of screenshots corresponding to each action taken. To assess a sorting task, the evaluator specifically examines the webpage screenshot **immediately after** the agent has performed the sort action. This allows them to see the full list of products as presented on the page and verify if the sorting order correctly matches the user's preference (e.g., "price: low to high"). We will clarify this procedural detail in the evaluation section.
>
> **4. Regarding the Scope of E-commerce Sites**
>
> Thank you for highlighting this. We confirm that this detail should be in the main paper.
>
> The current version of DeepShop is evaluated on **Amazon.com**. This was a deliberate "depth over breadth" design choice, allowing us to test agent capabilities in one of the world's most complex and widely-used e-commerce environments. We will make this scope explicit in the main body of our revised manuscript.

---

> ### Comment · Area_Chair_jC2p · 2025-11-25
> **Please participate in discussions with authors and other reviewers asap**
>
> Please ensure you are actively participating in the discussion phase.
>
> Additionally, I strongly encourage you to read the other reviews and discuss with your fellow reviewers. It is vital to compare perspectives and raise any remaining concerns now to give the authors a fair opportunity to respond.
>
> Based on these interactions, please update your reviews and finalize your decisions.
>
> Best, AC

---

### Official Review · Reviewer_33uR · 2025-11-02

**Soundness:** 2
**Presentation:** 2
**Contribution:** 2
**Rating:** 2
**Confidence:** 4

**Summary:**

This paper introduces DeepShop, an online benchmark for evaluating web shopping agents on complex, realistic shopping tasks. Starting from seed queries drawn from existing live-web benchmarks, the authors use LLM-driven diversity and complexity evolution to produce queries spanning five domains (Books, Electronics, Home, Fashion, Sports) and layered constraints (attributes, filters, sorting). Evaluation is both fine-grained (check attributes / filters / sorting separately) and holistic (task success only if all required aspects pass). Experiments on simple RAG, various web agents, and deep research systems show that all current methods struggle significantly, especially with applying filters and sorting preferences, resulting in low overall success rates and highlighting the benchmark's difficulty.

**Strengths:**

1. The paper identifies a significant limitation in existing web agent benchmarks. The focus on simple, deterministic tasks does not adequately test an agent's ability to handle the complex, multi-constraint queries that are common in real-world e-commerce. The proposed benchmark is on the right track of briging this gap of evaluating more advanced agent that has real-world application values.
2.  The proposed fine-grained evaluation framework is a key strength. It allows researchers to move beyond a simple binary success metric and diagnose why an agent failed (e.g., it found the right product attributes but failed to apply the sorting preference). This is well-demonstrated in the error analysis (Section 5.4) , which identifies critical failure points like grounding, replanning, and limited action spaces.

**Weaknesses:**

1. The paper lack crutial details on environment and reproducibility. While offline benchmarks are static and simpler, they offer a unique advantage in controllability and reproducibility over online enviroments. A core challenge of an online benchmark is the dynamic nature of websites (e.g., prices change, review counts increase, items go out of stock). The paper does not adequately address how the benchmark's ground truth is maintained. How can an agent's success on a query like "at least 300 reviews" or "check for the lowest price"  be reliably evaluated over time? This lack of clarity raises significant concerns about the benchmark's long-term utility and reproducibility. This has been a primiary issue that is discussed in previous work of online benchmarks (e.g. WebCanvas). However, the paper does not provide much details nor discussions about how this is managed in DeepShop.
2. The paper is vague about the specific e-commerce sites used. Screenshots and a prompt example suggest Amazon is a primary target, but this is not explicitly stated or confirmed as the only target. This may cause concerns in bias and robustness of the performance evaluation using the proposed benchmark.
3. The benchmark's focus on single-turn, complex queries may not be representative of realistic shopping scenariors, which is often an interactive, multi-turn process where preferences are refined (e.g., "Find me a laptop," "Okay, one for gaming," "Under $1000"). The paper's formulation as a single, complex instruction may be able to reflect the realistic distribution of user shopping intents.

**Questions:**

1. Given that DeepShop is an online benchmark, how does the evaluation framework handle the dynamic nature of live websites? How do the authors ensure that a task's ground truth remains stable for evaluation, and how can other researchers reliably reproduce the results months from now?
2. Could the authors please clarify which specific e-commerce websites are included in the DeepShop benchmark? Is it exclusively Amazon.com, or are other sites used?

---

> ### Author Response · Authors · 2025-11-23
> **Response to Reviewer 33uR**
>
> We sincerely thank the reviewer for their time and insightful feedback, which is instrumental in enhancing the quality of our work. We address the primary concerns below.
>
> **1. Regarding Environment Dynamicity and Reproducibility**
>
> Thank you for raising this critical point, which is a core challenge for all online benchmarks.
>
> Our evaluation framework is designed to address this by focusing on **relational invariance and procedural correctness**, rather than matching absolute, static ground truth values. For instance:
> *   For a query requiring sorting by "lowest price," we validate whether the agent correctly applies the sorting filter, not whether the resulting product price matches a specific value recorded months ago.
> *   Similarly, for a filter like "at least 300 reviews," the evaluation checks if the agent correctly filters for products that meet this criterion *at the time of execution*.
>
> This focus on procedural correctness ensures that we are evaluating the agent's reasoning and interaction capabilities, which is the primary goal of DeepShop. To ensure the long-term utility of the benchmark, we will also periodically verify the "solvability" of tasks to deprecate or update queries that become invalid due to significant website updates. We will add a detailed discussion of this evaluation philosophy to the paper to ensure clarity and aid reproducibility for future researchers.
>
> **2. Regarding the Vague Scope of E-commerce Sites**
>
> Thank you for the request for clarification. We acknowledge that this was not stated explicitly enough in the manuscript.
>
> The current version of the DeepShop benchmark is developed and evaluated primarily on **Amazon.com**. This was a deliberate design choice to prioritize **"depth over breadth."** Our goal was to first thoroughly test agent capabilities within a single, highly complex, and globally representative e-commerce environment before extending to others. This approach allows us to rigorously evaluate performance on deep navigation paths, a vast product space, and a wide array of complex UI elements. We will state this scope explicitly in the revised manuscript and have added the extension to other platforms (e.g., eBay, Walmart) as a key direction for future work in our conclusion.
>
> **3. Regarding the Representativeness of Single-Turn Queries**
>
> We appreciate this insightful comment on the nature of real-world shopping scenarios.
>
> While we agree that multi-turn interactions are a common and important scenario, single-turn complex queries represent an equally valid and highly challenging use case, particularly for users with clear, pre-defined goals. We position DeepShop as a benchmark that specifically tests an agent's ability to **decompose a complex, high-level user intent into a sequence of precise, low-level web interactions.**
>
> This decomposition capability is a **fundamental prerequisite** for any sophisticated conversational agent. Ultimately, a multi-turn dialogue converges to a final, complex set of constraints that must be executed faithfully—the very capability that DeepShop is designed to measure. We will revise our paper's introduction and task formulation to more clearly articulate this positioning.

---

> ### Comment · Area_Chair_jC2p · 2025-11-25
> **Please participate in discussions with authors and other reviewers asap**
>
> Please ensure you are actively participating in the discussion phase.
>
> Additionally, I strongly encourage you to read the other reviews and discuss with your fellow reviewers. It is vital to compare perspectives and raise any remaining concerns now to give the authors a fair opportunity to respond.
>
> Based on these interactions, please update your reviews and finalize your decisions.
>
> Best, AC

---

### Meta-Review · Area_Chair_pRXp · 2026-01-17

**Summary:**

This paper introduces a new benchmark for web shopping agents, which features live webpage navigation (on Amazon). The challenge of the dataset comes from diversity of interactions required (e.g. filtering, sorting, attribute understanding). All reviewers agree that this is a very relevant and realistic task for an agent benchmark.

One of the main concerns with the paper is its lack of generality: all tasks come from Amazon, so we don't actually test if these agents can truly generalize to arbitrary online marketplaces. While the authors explain that they wanted to focus on depth over breadth, multiple reviewers (and this AC) agree that a better version of this benchmark would target multiple platforms (even if not a huge number). The authors would be well served to at least do a small-scale generalizability test to other platforms.

Other major concerns include a lack of new insights in terms of error analysis: reviewers point out that the failure modes of shopping agents studied in this paper are similar to those in other web agent benchmarks. It would be nice to see something unique to this domain that is actionable (i.e., helps improve the next iteration of shopping agents specifically). Multiple reviewers were also concerned about the reproducibility of the benhcmark, as in how queries will be deprecated / added and what frequency checks will be done to ensure validity of the benchmark.

**Reviewer Concerns:**

Concerns addressed satisfactorily by rebuttal:
- specific question on how preferences about sorting are evaluated
- measurement of human performance
- how natural the queries are


Outstanding concerns:
- what do we actually learn from the error analysis compared to other web agent benchmark papers?
- focus on Amazon exclusively / lack of generalizability
- reproducibility and evolution of benchmark over time

**Reviewer Scores:**

33uR: unchanged at 2
znE7: unchanged at 2
gWjR: unchanged at 6
C7Vr: unchanged at 4

---

### Decision · Program_Chairs · 2026-01-26

Reject